# Synthesis and Electrochemical Performance of the Orthorhombic V_2_O_5_·*n*H_2_O Nanorods as Cathodes for Aqueous Zinc Batteries

**DOI:** 10.3390/nano12152530

**Published:** 2022-07-23

**Authors:** Xiaoping Tan, Gaoli Guo, Kaidi Wang, Huang Zhang

**Affiliations:** 1Research & Development Institute of Northwestern Polytechnical University in Shenzhen, Shenzhen 518057, China; iamxptan@mail.nwpu.edu.cn (X.T.); iamglguo@mail.nwpu.edu.cn (G.G.); iamkdwang@mail.nwpu.edu.cn (K.W.); 2Institute of Flexible Electronics, Northwestern Polytechnical University, Xi’an 710072, China; 3Ningbo Research Institute of Northwestern Polytechnical University, Ningbo 315103, China

**Keywords:** zinc batteries, V_2_O_5_·*n*H_2_O nanorods, cathode materials, Zn anode, aqueous batteries

## Abstract

Aqueous zinc-ion batteries offer the greatest promise as an alternative technology for low-cost and high-safety energy storage. However, the development of high-performance cathode materials and their compatibility with aqueous electrolytes are major obstacles to their practical applications. Herein, we report the synthesis of orthorhombic V_2_O_5_·*n*H_2_O nanorods as cathodes for aqueous zinc batteries. As a result, the electrode delivers a reversible capacity as high as 320 mAh g^−1^ at 1.0 A g^−1^ and long-term cycling stability in a wide window of 0.2 to 1.8 V using a mild ZnSO_4_ aqueous electrolyte. The superior performance can be attributed to the improved stability of materials, inhibited electrolyte decomposition and facilitated charge transfer kinetics of such materials for aqueous zinc storage. Furthermore, a full cell using microsized Zn powder as an anode within capacity-balancing design exhibits high capacity and stable cycling performance, proving the feasibility of these materials for practical application.

## 1. Introduction

With the fast development of renewable energy sources, there is an urgent need to develop low-cost, safe and sustainable energy storage devices to bridge the gap between intermittent energy generation and the distribution of power consumption [1,2]. Currently, lithium-ion batteries (LIBs) are dominating the market for powering portable electronics and electric vehicles, which is taking advantage of their high energy/power densities and efficiency [3]. However, volatile and flammable organic solvents in the electrolytes bring safety concerns [4]. Aqueous rechargeable batteries have emerged as a promising alternative technology, which benefit from the use of water-based electrolytes and similar working mechanisms making them especially competitive for large-scale energy storage [5,6].

Aqueous zinc-ion batteries (AZIBs) are one of the most promising systems due to their use of zinc metal as an anode, which has proper redox potential (−0.76 V vs. standard hydrogen electrode) for aqueous electrolytes, and high theoretical capacity (819 mAh g^−1^; 5851 mAh mL^−1^) [7,8,9,10,11]. However, the cathode materials capable of efficient Zn storage present the most critical challenge to achieving high-performance AZIBs. So far, only few cathode candidates have been reported for reversible Zn insertion/extraction, including manganese oxides, Refs. [12,13,14] Prussian blue analogues, Refs. [15,16] vanadium oxides, Refs. [17,18,19] and organic compounds [20,21,22].

Orthorhombic vanadium pentoxide (V_2_O_5_) holds great potential as a cathode material for AZIBs, owing to its high theoretical Zn-storage capacity of 589 mAh g^−1^ (based on the two-electron redox reaction of V as redox center) [23,24,25]. Its typically layered structure enables the efficient (de-)intercalation of metal ions, providing high capacity and rate capability [26,27,28]. However, its long-term cyclability is still unsatisfactory, due to the weak interactions of intercalated ions of Zn^2+^ with interlayers. Moreover, the dissolution of V-based compounds appears to be another issue in aqueous batteries, which affects cycling stability, especially in electrolytes with a low-pH value. To improve the redox kinetics and prolong the cycle life, various strategies have been reported. On one hand, by chemically pre-intercalated metal ions or foreign molecules, the layered vanadium oxides showed well-improved reversibility and cycling performance. In this respect, both alkali/alkaline-earth metal ions (Li^+^, [29] Na^+^, [30] Mg^2+^, [31] Ca^2+^, [32] etc.) and polar molecules (water, [33] pyridine, [34] etc.) have been adopted to pre-intercalate into the spaces between two bilayers, thus resulting in improved reaction kinetics and cycling stability of vanadium oxides. On the other hand, novel nanostructure design could be another approach to enabling efficient Zn-insertion in V_2_O_5_ for high-performance rechargeable aqueous Zn batteries [35,36,37,38,39].

Herein, we report the synthesis and electrochemical properties of orthorhombic V_2_O_5_·*n*H_2_O nanorods as cathode materials for AZIBs. In a mild 3 m (“m”: mol kg^−1^) ZnSO_4_ aqueous electrolyte, the electrochemical performance of a Zn/V_2_O_5_·*n*H_2_O cell was investigated in a wide voltage window from 0.2 to 1.8 V. Furthermore, full-cell configuration using microsized Zn powders as the anode is also reported, providing a realistic design of the Zn/V_2_O_5_ system for practical energy storage applications.

## 2. Materials and Methods

### 2.1. Material Synthesis

The V_2_O_5_·*n*H_2_O nanorods were synthesized using a typical hydrothermal method. Specifically, 0.6 g commercial V_2_O_5_ (98%, Alfa, China) was dispersed in 60 mL deionized water. After stirring for 30 min, 8 mL H_2_O_2_ (31wt%, Adamas, China) was added dropwise and vigorous stirring continued at 40 °C for 1 h. Then, the yellowish solution was transferred into a 100 mL Teflon-lined autoclave and kept at 200 °C for 48 h. After natural cooling, the powders were collected by centrifugation after being washed with DI water and ethanol. The final product was annealed at 250 °C for 2 h with a heating rate of 2 °C min^−1^ in air. Commercial V_2_O_5_ powders without any treatment and V_2_O_5_ nanorods annealed at 500 °C were used as a control.

### 2.2. Material Characterizations

The crystal phase of the product was characterized by X-ray diffraction (XRD) on a Bruker D8 Advance diffractometer (Cu Kα, Bruker, Karlsruhe, Germany). The morphology and microstructure were characterized by scanning electron microscopy (SEM, Gemini 300, ZEISS, Oberkochen, Germany) and transmission electron microscopy (TEM, FEI Tecnai G2 F30, 300 kV, Hillsboro, OR, USA). X-ray photoelectron spectrometer (XPS, ULVAC PHI Quantera, Chigasaki, Japan) was carried out to detect the surface chemical states of both pristine powders and cycled electrodes. The thermogravimetric analysis (TGA) was performed by a thermogravimetric analyzer (Netzsch, TGA-209F, Selb, Germany) at a ramping of 10 °C min^−1^.

### 2.3. Electrochemical Measurements

The electrochemical performance was investigated in CR2032-type coin cells. The cathode and Zn powder (99.99%, 600 mesh, Aladdin, Shanghai, China) electrodes were fabricated by doctor-blade casting slurries composed of 80 wt% active materials, 10 wt% carbon black (Super C65, Imerys) and 10 wt% polyvinylidene fluoride as binder (PVDF, Solef 6020, Solvay, Brussels, Belgium) in *N*-methyl-2-pyrrolidone (NMP, anhydrous, Innochem, Beijing, China) on stainless steel foil (type-316, thickness: 0.02 mm). After drying at 80 °C in air, electrodes of a diameter of 12 mm were punched and further dried at 12 °C under vacuum. The areal mass loading of V_2_O_5_ was about 1.5 mg cm^−2^. The areal mass loading of Zn powders was about 0.8–1.0 mg cm^−2^. The electrolyte was prepared by dissolving 3 mol ZnSO_4_·7H_2_O (99.5%, ACROS, Geel, Belgium) in 1 kg Millipore water (including the crystal water in the salt). Metallic zinc foil discs (0.03 mm thickness, Ø 16 mm, 99.95%, Goodfellow, Huntingdon, UK) and glass fiber discs (GF/D, Whatman, Ø 18 mm, Maidstone, UK) were punched as anodes and separators. Galvanostatic cycling was performed on an Automated Battery Test System (CT-4000, NEWARE, Shenzhen, China) in a voltage range from 0.2 to 1.8 V. The electrochemical workstation (CHI 660E, Shanghai, China) was used for the cyclic voltammetry (CV) and electrochemical impedance spectroscopy (EIS) tests. The EIS spectra were recorded in a frequency range from 100 kHz to 0.01 Hz with an AC amplitude of 5 mV at OCV states. The GITT test was performed to study the kinetics of cation diffusion in the V_2_O_5_ nanorod electrode using the following equation: D_Zn^2+^_ = (4/πτ)*[n_M_V_M_/S]^2^[ΔE_S_/ΔE_t_]^2^, in which the τ represents the relaxation time (s), n_M_ and V_M_ are the moles (mol) and molar volume (cm^−3^ mol^−1^) of V_2_O_5_, respectively, S is the electrode−electrolyte interface area (cm^2^), ΔE_S_ and ΔE_t_ are the steady-state voltage change and overall cell voltage after the application of a current pulse in a single step, respectively [35]. All the measurements were performed at room temperature (20 °C).

### 2.4. Differential Electrochemical Mass Spectrometry Measurement

The differential electrochemical mass spectrometry (DEMS) measurement was performed in an ECC-Air cell (EL-CELL, Hamburg, Germany). The cell was fabricated with a gas inlet and outlet and was connected to a bench-top gas mass spectrometer (Heiden HPR-20 QIC, Warrington, UK). The electrochemical measurement was controlled by a LAND CT2001A system (LAND, Wuhan, China). Upon the cell cycling, a constant stream of Ar (flow rate: 0.5 mL min^−1^) was maintained to take the gas products from the DEMS cell to the mass spectrometer.

## 3. Results

The V_2_O_5_·*n*H_2_O nanorods were prepared using a simple hydrothermal method followed by a moderate temperature annealing treatment. The illustration of the orthorhombic V_2_O_5_·*n*H_2_O nanorods is presented in Figure 1a, which visually demonstrates the designed nanorod structure. The crystal structure of orthorhombic V_2_O_5_·*n*H_2_O is built out of edge/corner-sharing VO5 tetragonal pyramids that form the layered structure [23]. The interlayer spacing can be intercalated by cations and molecules. The crystal phase of the nanorods was identified by XRD. As shown in Figure 1b, the powder XRD pattern of the as-synthesized V_2_O_5_·*n*H_2_O nanorods sample clearly exhibits characteristic peaks of the standard orthorhombic V_2_O_5_ phase (JCPDS no. 41-1426) with Pmmn space group, in which the peaks at 2θ 15.4°, 20.3°, 21.7°, 25.6°, 26.1°, 31.0°, 32.4° and 33.3°, correspond to the (200), (001), (101), (201), (110), (400), (301) and (011) planes, respectively [40]. The sharp and high-intensity diffraction peaks indicate the good crystallinity of the obtained materials.

To detect the surface chemical states on the nanorod materials, XPS analysis was performed. The high-resolution V 2p XPS spectra are shown in Figure 1c. The binding energies of the XPS spectra were calibrated using C 1s at 284.6 eV as a reference. The mainly deconvoluted V 2p_3/2_ and V 2p_1/2_ peaks at 517.5 and 525.0 eV can be attributed to the existence of V^5+^ in the orthorhombic V_2_O_5_·*n*H_2_O [40,41]. Small peaks were also detected at lower binding energies, indicating the existence of tetravalent vanadium (V^4+^) species [42,43]. This implies that the vanadium in the V_2_O_5_·*n*H_2_O nanorod surface layer was not completely oxidized when annealed at 250 °C, in accordance with the literature [37]. The O 1s spectrum (Appendix A) exhibits distinct peaks at around 532.6 and 531.2 eV, which can be attributed to the OH^−^ and pre-intercalated water molecules, while the main peak at 530.2 eV can be related to the vanadium oxide [44]. The existence of such a V-hydroxide-based interfacial structure is believed to enable the enhancement of interfacial behavior with facilitated electrochemical kinetics [45,46]. Thermogravimetric analysis was used to investigate the stability of the materials. As shown in Appendix A, the result shows a weight loss from room temperature to 370 °C under N_2_ flowing, which is related to the removal of pre-intercalated water and the decomposition of hydroxides. However, the sample exhibited a lower weight loss in the same temperature range under O_2_ flowing, and a slight weight increase was observed after 400 °C originating from the oxidization of vanadium to a higher valance state [37,47]. Figure 1d,e and Appendix A display the SEM images of the V_2_O_5_·*n*H_2_O nanorods. It can be seen that the synthesized V_2_O_5_·*n*H_2_O materials are composed of nanorods with a diameter of 30–300 nm and with various lengths. The commercial V_2_O_5_ powders are flake-shaped with an average size of ~1 um (Appendix A). The microstructure of the V_2_O_5_·*n*H_2_O nanorods is further confirmed by transmission electron microscopy (TEM) images (Appendix A and Figure 1f). The results clearly demonstrate the typically one-dimensional structure of the materials. The fast Fourier transform (FFT) profile (Figure 1f inset) and the high-resolution TEM image (Figure 1g) clearly exhibit lattice fringes of 0.34 nm, which refer to the (110) planes of orthorhombic V_2_O_5_·*n*H_2_O. The EDX results indicate the presence of V and O elements in the synthesized V_2_O_5_·*n*H_2_O nanorods with an atomic ratio of 0.41, which is close to the theoretical value of V_2_O_5_ of 0.40 (Appendix A).

To investigate the Zn-storage properties of the V_2_O_5_·*n*H_2_O nanorods, coin cells were fabricated using zinc foil as the anode in 3 m ZnSO_4_ aqueous electrolytes. Figure 2a shows the cyclic voltammetry (CV) plots at a scan rate of 3 mV s^−1^ in a voltage window of 0.2–1.8 V. Upon the initial cathodic scan, there was a broad reduction peak at ~0.9 V, and a weak peak at ~0.6 V, indicating the stepwise insertion of Zn^2+^ into the V_2_O_5_·*n*H_2_O nanorods [23]. During the anodic scan, two oxidation peaks at 1.16 and 1.28 V were observed due to the extraction of Zn from the host. In the deep cycles, all the peaks did not shift, however, the currents increased slightly, which can be attributed to the structural activation of the materials due to a gradual insertion of Zn^2+^. Appendix A shows the CV curves of the V_2_O_5_·*n*H_2_O nanorod electrode at various scan rates from 0.5 to 5 mV s^−1^. As the scan rate increases, the curves retain a similar shape, indicating the fast redox reaction kinetics of these nanorod materials for Zn batteries. It should be mentioned that the working voltage window applied for CV measurement was larger than that of most reported vanadium-based cathodes in either Zn (CF_3_SO_3_)_2_ or ZnSO_4_ electrolytes [19,23,44], there was no obvious oxygen evolution reaction detected upon the anodic scan at 1.8 V, which shows the unique electrochemical stability of such materials in aqueous ZnSO_4_ electrolyte.

The galvanostatic charge–discharge (GCD) test was performed under different current densities from 0.1 to 5.0 A g^−1^ in a voltage range of 0.2–1.8 V (Figure 2b). The voltage profiles of the V_2_O_5_·*n*H_2_O nanorods at a low-current density of 0.1 A g^−1^ exhibit distinct plateaus at around 0.9 V and 0.6 V, in agreement with the CV results. Even when increasing the current density to 2.0 A g^−1^ the plateaus still remained, indicating a good kinetic reaction of the electrode materials. The specific capacities of the electrode at various current densities from 0.05 A g^−1^ to 5.0 A g^−1^ for five cycles are presented in Figure 2c. The electrode can deliver reversible capacities of 540.4, 410.2, 390.2, 364.5, 320.1, 297.3 and 197.7 mAh g^−1^ at 0.05, 0.1, 0.2, 0.5, 1.0, 2.0 and 5.0 A g^−1^, respectively. When the current returns to lower values, the capacities can almost be recovered. It should be noted that capacity fading and low Coulombic efficiency were commonly observed at low current density, which can be attributed to the inevitable side reaction of the electrode materials with the aqueous electrolyte [48]. A long-term cycling test was also performed at 1.0 A g^−1^ with three initial cycles at low current density as an activation step. As seen in Figure 2d, the capacity can maintain at 230.3 mAh g^−1^ after 500 cycles with an average Coulombic efficiency of ~99.7%. The selected GCD profiles can be found in Appendix A. For comparison, the performance of commercial V_2_O_5_ (Appendix A) was also investigated, and the electrode only delivered a capacity of 130 mAh g^−1^ at 1.0 A g^−1^, which is much lower than that of the V_2_O_5_·*n*H_2_O nanorod. Moreover, the V_2_O_5_ nanorod sample annealed at 500 °C, which removed the intercalated water, showed poor cycling stability with low specific capacity, as displayed in Appendix A. Note that the capacities of commercial V_2_O_5_ and V_2_O_5_ nanorods annealed at a higher temperature increased in the initial cycles, which can be attributed to the transformation of V_2_O_5_ into hydrated V_2_O_5_ [49]. However, the nanorod structure annealed at a moderate temperature (250 °C) enabled superior high-rate capability of orthorhombic V_2_O_5_ and long-term cycling stability in aqueous Zn batteries.

On the basis of the CV data at various scan rates, the electrochemical kinetic process can be determined by the following power–law relationship:*i* = a*v*^b^
where *i* and *v* denote the response current and scan rate, respectively, and a and ^b^ are both adjustable parameters. Figure 3a shows the b values of peaks 1–4 to be 0.53, 0.66, 0.72, and 0.62. A b value of 0.5 indicates that the current is diffusion-controlled. The achieved results imply that the corresponding redox reactions of the V_2_O_5_ nanorod electrode at the peak regions are mainly limited by ion diffusion within the scan rates ranging from 0.5 to 5 mV s^−1^. To separate the capacitive contribution, the equation *i*(*v*) = k_1_*v* + k_2_*v*^1/2^ was used to distinguish the ion intercalation part (k_1_*v*) and the capacitance part (k_2_*v*^1/2^), as shown in Figure 3b. At 5 mV s^−1^, 41.7% of the total current (capacity) is capacitive, indicating that the diffusion contribution holds the dominant position in the total capacity, and the corresponding reaction is mainly limited by the Zn^2+^ ion diffusion rate.

In order to obtain the Zn^2+^ ion diffusion coefficient of V_2_O_5_ nanorod electrodes, GITT tests were carried out during the second cycle (Figure 3c). The corresponding diffusion coefficient of Zn^2+^ (D_Zn_) during the second cycle was calculated to estimate the kinetics of V_2_O_5_·*n*H_2_O nanorods, which was in the range of ca. 10^−9^–10^−11^ cm^2^ s^−1^ (Figure 3d) and was higher than the that of bulk V_2_O_5_ [39]. The Zn^2+^ ion diffusivity on discharging at the last stage was lower than the average value due to the Zn^2+^ ion enriched in the cathode. Overall, the V_2_O_5_·*n*H_2_O nanorods showed a high diffusion coefficient, demonstrating the rapid diffusion kinetics of Zn^2+^ in the materials.

To understand the structural evolution and surface chemical change of the electrode material, XRD and XPS measurements were applied to the electrodes at different charge and discharge states. Figure 4a shows the typical charge–discharge curve of the V_2_O_5_·*n*H_2_O nanorod electrode, indicating the selected states for the measurements. The XRD profiles at open-circuit voltage (OCV), different depth of discharge and state of charge states are presented in Figure 4b. The diffraction peak of (001) plane at 26.3° shifts to higher angles during the discharge process from OCV to 0.8 V and a new peak at 13.6° appears, indicating that the Zn intercalation leads to the formation of a new phase with larger interlayer spacing [36,50]. Upon deep discharge to 0.2 V, the new peaks at 24.1° and 28.2° were clearly observed, which can be attributed to the ZnV_3_O_8_ (JCPDS no. 24-1481) and Zn_2_V_2_O_7_ (JCPDS no. 52-1893) phases. This phenomenon with new phases resulting from Zn insertion in vanadium-based oxides cathodes is commonly reported in the literature [18,33,48]. As reported, the Zn-based precipitation, such as the ZnSO_4_[Zn(OH)_2_]_3_·*x*H_2_O (ZHS) phase, is widely detected when using ZnSO_4_ aqueous electrolyte, however, the signal of the ZHS phase was not observed in our case possibly due to the fact that the concentration of OH^−^ is not high enough in the electrolyte to generate large amounts of ZHS [51,52]. When the electrode was charged to 1.1 V, the diffraction peak of (001) plane moves back to its original position, indicating the materials’ stability after Zn extraction. At the fully charged state, the main peaks of the orthorhombic V_2_O_5_ phase can be fully recovered. Thus, from the XRD results, we observed highly reversible Zn insertion with induced zinc vanadium oxides in the V_2_O_5_·*n*H_2_O nanorod materials alongside new phase changes.

Focusing on the surface valence states of vanadium during the Zn^2+^ intercalation process, the V 2p XPS spectra of V_2_O_5_·*n*H_2_O nanorods at different electrochemical stages are shown in Figure 4c and Appendix A. Similar to the pristine material, the fresh electrode exhibits two V 2p_3/2_ peaks at 517.5 eV and 516.8 eV, which can be assigned to the V^5+^ and V^4+^ species. The analysis of the discharged cathode revealed a third contribution at 515.8 eV that corresponds to V^3+^ species, along with an increase in V^4+^ and a decrease in V^5+^ as a result of Zn^2+^ intercalation and the consequent reduction of V_2_O_5_ framework. During the charging process, the V is oxidized from V^3+^ and V^4+^ to V^5+^ due to the simultaneous Zn extraction. It is worth mentioning that there is still trace amount of V^3+^ remaining in the fully charged electrode, which can be attributed to the incomplete extraction of Zn^2+^. On the basis of the above electrochemical results, we propose a Zn-storage mechanism in Figure 4d, showing the typical (de-)intercalation behaviors upon the discharge–charge processes of the V_2_O_5_·*n*H_2_O materials.

SEM was conducted on the electrode after 200 cycles to investigate the morphology changes after the long-term Zn (de-)intercalation process. The SEM images (Appendix A and Figure 5a) demonstrate that there is no obvious morphology change and the nanorod structure almost remains. Such a good material stability ensures the cycling stability of the V_2_O_5_·*n*H_2_O nanorod material for aqueous Zn-storage. To evaluate the interfacial behaviors, Nyquist plots, consisting of a semicircle associated to the charge transfer at electrode–electrolyte interface and a slop line reflecting the ion diffusion in the bulk electrode, were collected on the commercial V_2_O_5_ powder and V_2_O_5_·*n*H_2_O nanorod electrodes before and after 200 cycles (Figure 5b) [35]. The commercial V_2_O_5_ exhibits a significant increase in the charge-transfer resistance after cycling, while the V_2_O_5_·*n*H_2_O nanorod only possesses an increase from ~40 Ω to ~140 Ω, suggesting that the designed nanostructure with a hydroxide-based interfacial structure can efficiently stabilize the electrode materials and facilitate charge transfer at the interface, consequently prolonging the cycling stability and reversibility upon Zn insertion.

Although cycling in a wide window can result in higher energy, the electrolyte decomposition generated from the water electrolysis gets more stress. In situ analysis of electrolyte compatibility with electrodes (both Zn and V_2_O_5_) was conducted by DEMS, to detect the gas evolution of the cell upon cycling. The mass spectrometric current traces for O_2_ (*m*/*z* = 32) and H_2_ (*m*/*z* = 2) are presented in Figure 5c. There was no O_2_ evolution detected upon the initial one and a half cycles. The electrochemically induced H_2_ peak only occurred when charged to 1.8 V, while the current intensity was rather low (only ~0.35 pA) [53].

Up to now, most of the research on aqueous Zn batteries has directly used Zn foil as the anode, which is not feasible for real industrial application due to its unmatching to current electrode manufacturing technologies [54]. Therefore, a Zn/V_2_O_5_·*n*H_2_O battery using microsized Zn powders as the anode was also fabricated. Such a technology promises capacity balance design in a full battery. Figure 6 shows the electrochemical performance of the Zn (powder)/V_2_O_5_·*n*H_2_O cell in the voltage range from 0.2 to 1.8 V. This cell is cathode-limited in a cathode to anode capacity ratio of 1:1.2 (according to the initial capacity of V_2_O_5_ and theoretical capacity of Zn). As shown in Figure 6a, the cell can deliver a reversible capacity of ~500 mAh g^−1^ at 0.05 A g^−1^, and maintain at 330 mAh g^−1^ at 1.0 A g^−1^, demonstrating similar capacities to the cells using Zn foil as the anode. Based on the active materials in both electrodes, the energy density of such cells reaches 255 Wh Kg^−1^. The rate performance is presented in Figure 6b. As seen, the cell can deliver capacities of 415.6, 379.7, 350.0, 316.3, 280.3 and 184.9 mAh g^−1^ at 0.1, 0.2, 0.5, 1.0, 2.0 and 5.0 A g^−1^, respectively, and the capacity can recover to 270.0 mAh g^−1^ when the current decreases to 2.0 A g^−1^. It should be noted that the capacity decay was faster than that of a half-cell and the capacities at higher current densities were slightly lower than those of cells using zinc foils as anodes, which may result from vanadium dissolution and the severe side reaction of zinc powders in aqueous electrolytes and can be further improved by structural modification and electrolyte optimization [54,55,56,57]. The cell was also cycled at 1.0 A g^−1^ for 100 cycles (Figure 6c). The capacity can retain at 157 mAh g^−1^ after cycling with high CEs of ~98.9%, The electrochemical performance of reported orthorhombic V_2_O_5_ materials in aqueous Zn batteries are summarized in Appendix A, apparently proving the superior cycle life, high efficiency and high-capacity utilization not only in cells using metallic Zn foil as the anode but also in realistic configuration.

## 4. Conclusions

In summary, high-performance orthorhombic V_2_O_5_·*n*H_2_O nanorod materials have been synthesized with mixed vanadium valences, a hydroxide surface layer and pre-intercalated water characteristics. As cathodes for aqueous zinc batteries in a 3 m ZnSO_4_ aqueous electrolyte, the obtained materials show superior electrochemical performance in terms of high specific capacity, power capability and cycling stability. By using both structural analysis and electrochemical characterizations, it was proven that the unique nanostructure and interfacial behavior significantly improved the material stability of orthorhombic V_2_O_5_ in aqueous media, suppressed the electrolyte decomposition at high operating voltage, and facilitated charge transfer kinetics. Furthermore, a realistic full cell using Zn powder as the anode with capacity balance design was also reported to deliver excellent capacity utilization and cycling stability. This work provides a strategy for enabling orthorhombic V_2_O_5_ materials to be efficient in aqueous Zn batteries and proves the feasibility of such materials for real industrial applications.

## Figures and Tables

**Figure 1 nanomaterials-12-02530-f001:**
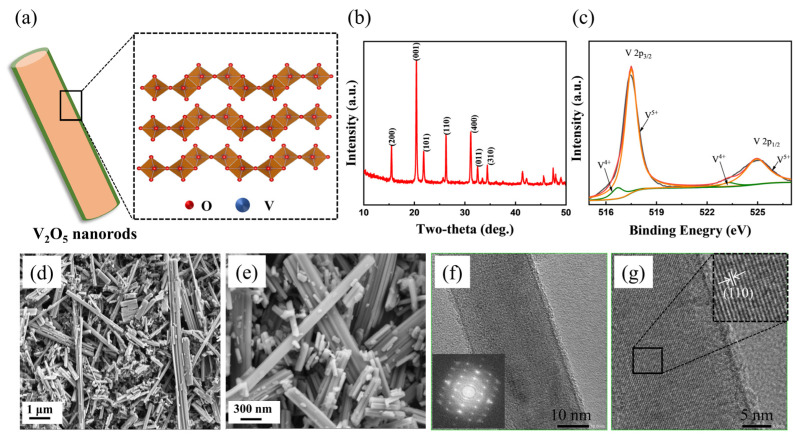
(**a**) Schematic illustration of the orthorhombic V_2_O_5_ nanorod structure. XRD pattern (**b**), V 2p XPS spectrum (**c**), SEM images (**d**,**e**) and TEM images (**f**,**g**) of the synthesized V_2_O_5_ materials.

**Figure 2 nanomaterials-12-02530-f002:**
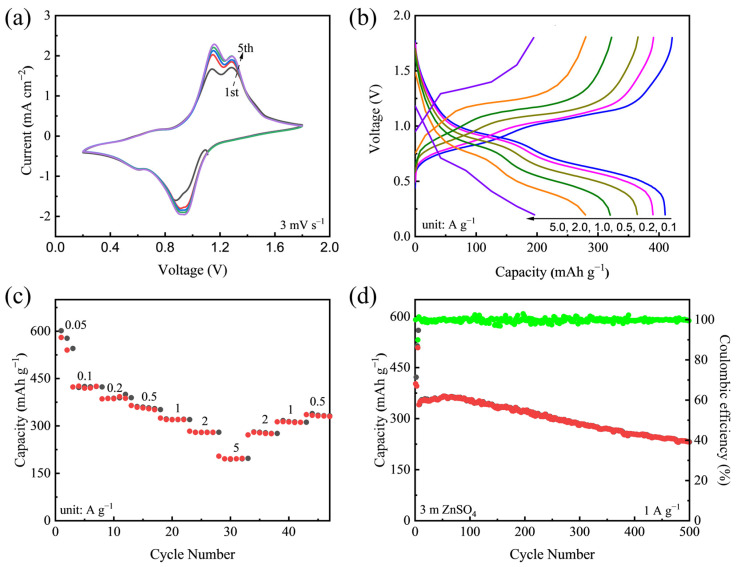
(**a**) CV curves of the V_2_O_5_·*n*H_2_O nanorod electrode in a 3 m ZnSO_4_ aqueous electrolyte using Zn foil as a counter electrode at a scan rate of 3 mV s^−1^. (**b**) Galvanostatic discharge–charge (GCD) voltage profiles of the V_2_O_5_·*n*H_2_O nanorod electrode at current densities from 0.1 to 5.0 A g^−1^. (**c**) Specific capacities of the V_2_O_5_·*n*H_2_O nanorod electrode at various current densities for 5 cycles. (**d**) Long-term cycling performance and corresponding Coulombic efficiencies of the V_2_O_5_·*n*H_2_O nanorod electrode at 1.0 A g^−1^.

**Figure 3 nanomaterials-12-02530-f003:**
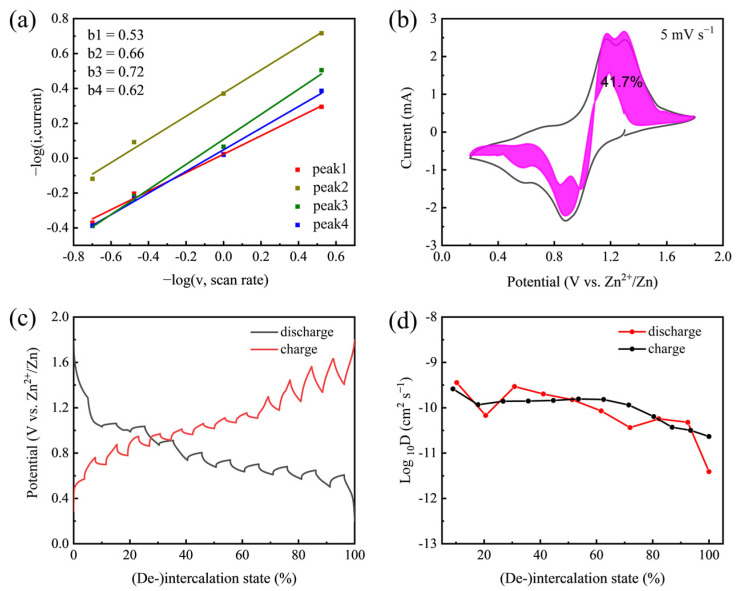
(**a**) The plots of–log (*i*) versus–log (*v*) of cathodic and anodic peaks. (**b**) Capacitive contribution (pink part) and diffusion-controlled contribution (void part) at 5 mV s^−1^. (**c**) Discharge–charge curves of V_2_O_5_·*n*H_2_O nanorods in GITT measurement. (**d**) The diffusion coefficients of Zn^2+^ upon the 2nd discharge and charge progresses of the V_2_O_5_·*n*H_2_O nanorods.

**Figure 4 nanomaterials-12-02530-f004:**
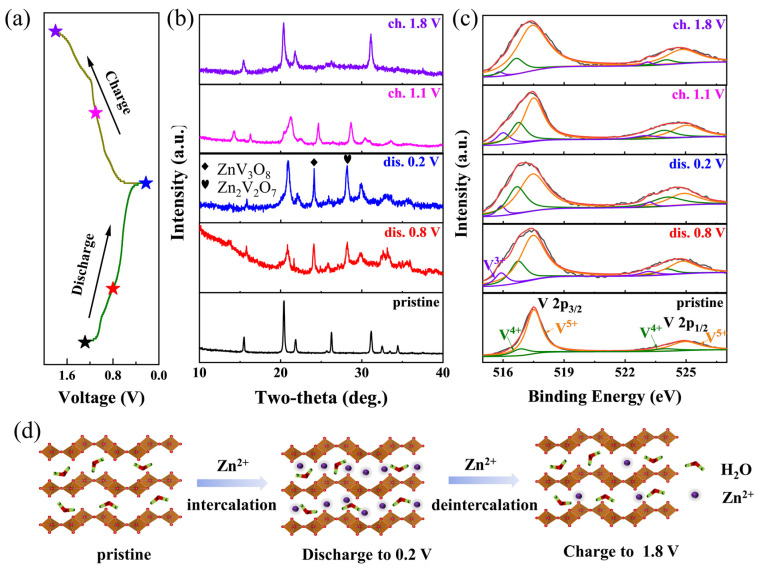
(**a**) Typical galvanostatic discharge–charge (GCD) profiles of the V_2_O_5_ nanorods in aqueous Zn batteries. (**b**) XRD patterns and (**c**) V 2p XPS spectra of the V_2_O_5_ nanorod materials at different discharge and charge states. (**d**) The schematic of ion storage mechanism upon discharging and charging.

**Figure 5 nanomaterials-12-02530-f005:**
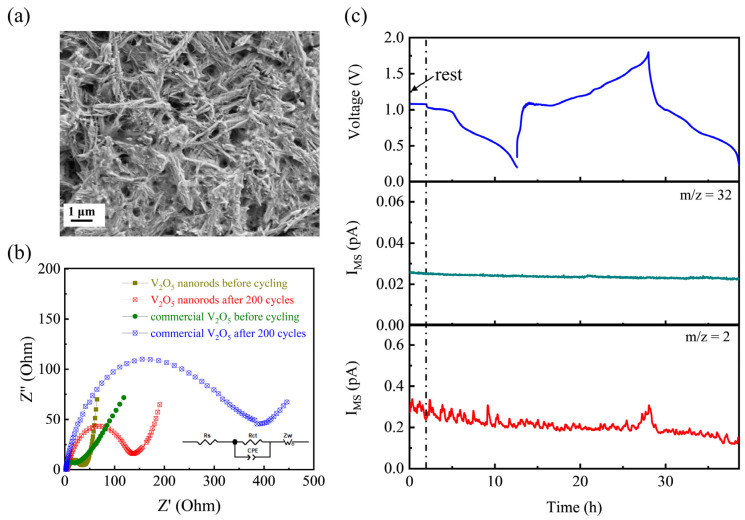
(**a**) SEM image of the cycled electrode after 200 cycles. (**b**) Nyquist plots of the commercial V_2_O_5_ powder and V_2_O_5_·*n*H_2_O nanorod electrodes before and after cycling. (**c**) Online DEMS data for Zn/ V_2_O_5_·*n*H_2_O cell in 3 m ZnSO_4_ aqueous electrolyte in the voltage window of 0.2–1.8 V.

**Figure 6 nanomaterials-12-02530-f006:**
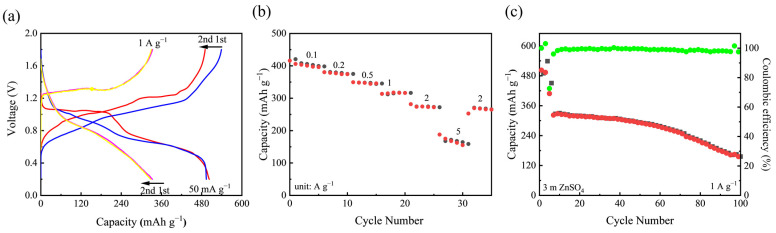
(**a**) GCD profiles of the Zn/V_2_O_5_·*n*H_2_O cell using Zn powders as the anode at 50 mA g^−1^ and 1 A g^−1^ for the 1st and 2nd cycles. (**b**) Rate performance of the Zn/V_2_O_5_·*n*H_2_O cell at current densities from 0.1 to 5.0 A g^−1^. (**c**) Cycling performance of the cell at 1.0 A g^−1^. The cell was initially cycled at 0.05 A g^−1^ for three cycles as an activation step.

## Data Availability

Not applicable.

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
