# Peer review of "Synthesis and Electrochemical Performance of the Orthorhombic V2O5·nH2O Nanorods as Cathodes for Aqueous Zinc Batteries"

_nanomaterials, 2022, doi:10.3390/nano12152530_

Round 1

Reviewer 1 Report

I believe that the paper entitled "Synthesis and electrochemical performance of the orthorhombic V2O5·nH2O nanorods as cathodes for aqueous zinc batteries" is suitable for publication in Nanomaterials after the authors consider the following minor points:

1. Please include the following reference along with [7-10]

Nanomaterials 11 (2021) 656

2. The authors need to compare the performance of the materials and coin cells evaluated in this paper with the literature.

Author Response

Response to Reviewer 1:

Comments and Suggestions for Authors

I believe that the paper entitled "Synthesis and electrochemical performance of the orthorhombic V2O5·nH2O nanorods as cathodes for aqueous zinc batteries" is suitable for publication in Nanomaterials after the authors consider the following minor points:

Response: Thanks a lot for the highly recommendation from the reviewer. The related revisions have been made to improve the manuscript.

  1. Please include the following reference along with [7-10]

Nanomaterials 11 (2021) 656

Response: The reference has been added in the revised manuscript.

  1. The authors need to compare the performance of the materials and coin cells evaluated in this paper with the literature.

Response: Thanks a lot for the suggestion. We have added the performance comparison in the Table S1 (Supporting Information).

Reviewer 2 Report

This work reports on the synthesis of orthorhombic V2O5·nH2O nanorods cathodes for aqueous zinc ion batteries. As a result, the electrode delivers a reversible capacity as high as 320 mAh g-1 at 1.0 A g-1  and long-term cycling stability in a wide window of 0.2 to 1.8 V using a mild ZnSO4 aqueous electrolyte. The superior performance is attributed to the improved materials stability, inhibited electrolyte decomposition and facilitated charge transfer kinetics of such materials for aqueous zinc storage.

This a well-structured manuscript with very interesting results that can be published after following major revisions.

It is well-known that V2O5 is prone to dissolution especially at very low pH values that might lead to HSO4-/H2SO4 formation in ZnSO4 solution (see Pourbaix diagram). This very important aspect related to chemical/electrochemical stability of V2O5 material is completely missing in that work and must be discussed into details. Hao et al. (DOI: 10.1002/cnma.202000105) reports on a color change of the V2O5/ 3 M ZnSO4 electrolyte from transparent to yellow after only one day. The huge decay in ZIB cell performance after only 100 cycles shown in Fig. 6c might be an indication for such an irreversible process.

Line 179. “there is no obvious hydrogen evolution reaction detected upon the anodic scan to 1.8 V, showing the unique electrochemical stability of such materials in aqueous ZnSO4 electrolyte”. At this potential value, I wouldn´t expect a H2 but an O2 evolution. It seems that there is some confusion between hal-cell and full-cell configuration in Fig. 3b. The axis label shoulb be "potential" instead "voltage" and "vs. reference electrode" has to be added.

Line 183 Fig 3b) “Capacitive contribution (pink part) and diffusion-controlled contribution (void part) at 5 mV s-1.”  This statement is not clear and should be clarified and commented more in details. Do the authors mean intercalation/deintercalation-induced pseudo-capacitive charge? Why diffusion-controlled? By using oxide materials with large interlayer spacing like V2O5, one would expect also kinetic-controlled behavior/region as well that is responsible for relatively high current densities.     

Author Response

Response to Reviewer 2:

Comments and Suggestions for Authors

This work reports on the synthesis of orthorhombic V2O5·nH2O nanorods cathodes for aqueous zinc ion batteries. As a result, the electrode delivers a reversible capacity as high as 320 mAh g-1 at 1.0 A g-1 and long-term cycling stability in a wide window of 0.2 to 1.8 V using a mild ZnSO4 aqueous electrolyte. The superior performance is attributed to the improved materials stability, inhibited electrolyte decomposition and facilitated charge transfer kinetics of such materials for aqueous zinc storage.

This a well-structured manuscript with very interesting results that can be published after following major revisions.

Response: Thanks a lot for the highly recommendation from the reviewer. The related revisions have been made to improve the manuscript.

  1. It is well-known that V2O5 is prone to dissolution especially at very low pH values that might lead to HSO4-/H2SO4 formation in ZnSO4 solution (see Pourbaix diagram). This very important aspect related to chemical/electrochemical stability of V2O5 material is completely missing in that work and must be discussed into details. Hao et al. (DOI: 10.1002/cnma.202000105) reports on a color change of the V2O5/ 3 M ZnSO4 electrolyte from transparent to yellow after only one day. The huge decay in ZIB cell performance after only 100 cycles shown in Fig. 6c might be an indication for such an irreversible process.

Response: Thanks a lot for the suggestion. In aqueous batteries, the V-based compounds suffer from the similar dissolution issues, especially at low pH electrolyte, leading to the inevitable capacity decay. In the full cell, the zinc powders were used as anode in a cathode-limited design. The capacity decay should be attributed to the vanadium dissolution, and severe side reaction of zinc powders in the aqueous electrolytes, leading to the faster capacity decay than that in half-cell. [ChemNanoMat 2020, 6, 797-805; Advanced Energy Materials 2021, 11, 202003931] The related discussion has been added in the revised manuscript.

  1. Line 179. “there is no obvious hydrogen evolution reaction detected upon the anodic scan to 1.8 V, showing the unique electrochemical stability of such materials in aqueous ZnSO4 electrolyte”. At this potential value, I wouldn´t expect a H2 but an O2 evolution. It seems that there is some confusion between half-cell and full-cell configuration in Fig. 3b. The axis label should be "potential" instead "voltage" and "vs. reference electrode" has to be added.

Response: Thanks a lot for pointing out the mistake. It should be oxygen evolution upon anodic scan to 1.8 V and the description has been corrected. The figure 3 has been refined.

  1. Line 183 Fig 3b) “Capacitive contribution (pink part) and diffusion-controlled contribution (void part) at 5 mV s-1.” This statement is not clear and should be clarified and commented more in details. Do the authors mean intercalation/deintercalation-induced pseudo-capacitive charge? Why diffusion-controlled? By using oxide materials with large interlayer spacing like V2O5, one would expect also kinetic-controlled behavior/region as well that is responsible for relatively high current densities.

Response: Thank the reviewer for his/her questions. The diffusion-controlled process refers to the solid state Zn2+ ion diffusion in the bulk electrode. The capacitive charge storage can be attributed to the rapid redox reaction on the surface of the cathode. At 5 mV s-1, 41.7% of the total current (capacity) is capacitive, showing that the diffusion contribution holds the dominant position in the total capacity and the corresponding reaction is mainly limited by the Zn2+ ion diffusion rate. This related discussion has been implemented in the revised manuscript.

Round 2

Reviewer 2 Report

Since all my comments/remarks have been considered, the manuscript can be published in that form